# Endothelial Biomarkers Are Superior to Classic Inflammatory Biomarkers in Community-Acquired Pneumonia

**DOI:** 10.3390/biomedicines12102413

**Published:** 2024-10-21

**Authors:** Paula González-Jiménez, Mónica Piqueras, Ana Latorre, Jordi Tortosa-Carreres, Noé Mengot, Ricardo Alonso, Soledad Reyes, Isabel Amara-Elori, Luis Martínez-Dolz, Antonio Moscardó, Rosario Menéndez, Raúl Méndez

**Affiliations:** 1Pneumology Department, La Fe University and Polytechnic Hospital, 46026 Valencia, Spain; 2Respiratory Infections, Health Research Institute La Fe (IISLAFE), 46026 Valencia, Spain; 3Medicine Department, University of Valencia, 46010 Valencia, Spain; 4Laboratory Department, La Fe University and Polytechnic Hospital, 46026 Valencia, Spain; 5Cardiology Department, La Fe University and Polytechnic Hospital, 46026 Valencia, Spain; 6Centre for Biomedical Research Network in Cardiovascular Diseases (CIBERCV), Instituto de Salud Carlos III, 28029 Madrid, Spain; 7Haemostasis and Thrombosis Unit, La Fe University and Polytechnic Hospital, 46026 Valencia, Spain; 8Centre for Biomedical Research Network in Respiratory Diseases (CIBERES), Instituto de Salud Carlos III, 28029 Madrid, Spain

**Keywords:** CAP, MR-proADM, CT-proET-1, inflammatory biomarkers, cardiovascular events

## Abstract

**Background:** Complications in community-acquired pneumonia (CAP), including cardiovascular events (CVE), can occur during an acute episode and in the long term. We aimed to analyse the role of endothelial damage biomarkers (C-terminal endothelin-1 precursor fragment [CT-proET-1] and mid-regional pro-adrenomedullin [MR-proADM]), in contrast to classic inflammation markers (C Reactive Protein [CRP] and procalcitonin [PCT]) in patients admitted for CAP and their relationship with ICU admission, CVE and mortality in the short and long term; **Methods:** Biomarkers were analysed in 515 patients with CAP at day 1, 285 at day 5 and 280 at day 30. Traditional inflammatory biomarkers and endothelial damage biomarkers were measured. ICU admission, CVE and mortality (in-hospital and 1-year follow-up) were assessed using receiver operating characteristic (ROC) curve analysis and univariate logistic regression. **Results:** A statistically significant association was observed between initial, raised CT-proET-1 and MR-proADM levels, the need for ICU admission and the development of in-hospital CVE or in-hospital mortality. Both endothelial markers maintained a strong association at day 30 with 1-year follow-up CVE. At day 1, CRP and PCT were only associated with ICU admission. On day 30, there was no association between inflammatory markers and long-term CVE or death. The odds ratio (OR) and area under the curve (AUC) of endothelial biomarkers were superior to those of classic biomarkers for all outcomes considered. **Conclusions:** Endothelial biomarkers are better indicators than classic ones in predicting worse outcomes in both the short and long term, especially CVE. MR-proADM is the best biomarker for predicting complications in CAP.

## 1. Introduction

Community-acquired pneumonia (CAP) poses a considerable global burden in terms of mortality, morbidity, and economic costs [1]. Complications associated with CAP can manifest during the acute phase and beyond, including the occurrence of acute cardiovascular events (CVE) [2]. Cases of myocardial infarction, heart failure, arrhythmias, and stroke have been observed as rising following an episode of CAP even years later [3]. The implications of this association at a population level are noteworthy [4,5].

Procalcitonin (PCT) is a prohormone primarily produced in the C cells of the thyroid gland. Minor production of PCT also occurs in the neuroendocrine tissue of other organs. Whilst its concentration is typically low in healthy individuals, it increases during bacterial infections. In addition, during such infections, cells synthesise PCT at the place of the infection. Several studies have shown that PCT levels have risen in the presence of bacteraemia and severe infections, serving as a predictor of mortality in patients with CAP and sepsis [6].

C-reactive protein (CRP) is a widely used biomarker for inflammation in diverse pathologies. The liver promptly elevates the synthesis of CRP in response to cytokines released at the site of pathology. As a result, CRP levels serve to indicate the intensity of an inflammatory response [7]. However, its elevation kinetics may result in relatively lower values in the very early stages [8]. Moreover, its low specificity is noteworthy, given that it increases in various pathologies. Some studies have shown that lower CRP levels are an independent predictor for the absence of severe complications in CAP [9,10].

Mid-regional pro-adrenomedullin (MR-proADM) is a stable and identifiable 48-amino acid fragment derived from adrenomedullin (ADM). It belongs to the calcitonin family [11]. Methodologically, the measurement of ADM is not feasible due to its short half-life, 22 min. Therefore, MR-proADM is measured instead. This is due to the fact that it is in a 1:1 ratio with ADM and is chemically stable, thus allowing for a reliable measurement of its concentration [12,13,14]. Part of its synthesis takes place in the vascular endothelium, and its principal physiologic effect is vasodilation alongside bronchodilatation, cardiac contractility and glomerular filtration [15]. Recent studies have noted its increase with different pathologies such as cardiac disease, respiratory infections and sepsis [16].

Endothelin-1 (ET-1) is primarily synthesised by the vascular endothelium and plays a crucial role in maintaining vascular tone homeostasis. Increased levels of ET-1 have been linked to vasoconstriction, vascular hypertrophy, inflammation and pulmonary fibrosis [17]. The C-terminal endothelin-1 precursor fragment (CT-proET-1) measurement serves as a method to indirectly evaluate the release of ET-1. This molecule is present in human plasma and exhibits greater stability compared to vasoconstrictor ET-1. Both CT-proET-1 and MR-proADM are associated with in-hospital mortality, in-hospital CVE and 1-year mortality since admission for either CAP or SARS-CoV-2 pneumonia [18].

Thus, the aim of the study is to examine the usefulness of endothelial damage biomarkers (CT-proET-1 and MR-proADM) compared to inflammation biomarkers (CRP and PCT) in predicting the need for Intensive Care Unit (ICU) admission, CVE (in-hospital and 1-year follow-up) and mortality (in-hospital and 1-year follow-up), in patients admitted for CAP.

## 2. Materials and Methods

### 2.1. Study Design and Study Population

We conducted a prospective, observational study in hospitalised patients with CAP at La Fe University and Polytechnic Hospital in Valencia (Spain). The Biomedical Research Ethics Committee Hospital La Fe approved the study (2013/0204).

A diagnosis of pneumonia requires compatible signs and symptoms and a new radiologic infiltration. Exclusion criteria included residing in a nursing home, being under 18 years old, being admitted in the previous 15 days, having immunosuppression, and refusing written informed consent.

Relevant demographics, comorbidities, analytical parameters, and microbiologic and radiographic data were recorded. Initial severity was assessed using the Pneumonia Severity Index (PSI) score and SpO2/FiO2 measurement at the emergency department (ED). PSI score was dichotomised by merging the groups into classes 1–3 (low-moderate risk) and classes 4–5 (high risk). All hospitalised patients were monitored during 1-year follow-ups or until death if it had occurred previously.

### 2.2. Definition of Clinical Outcomes

In-hospital outcomes considered were the development of a CVE, the need for ICU admission (both directly or during hospital stay) and death. One-year follow-up complications were considered, such as the occurrence of a CVE or death.

CVE was considered if acute coronary syndrome, new or worsening congestive heart failure, new or recurrent arrhythmia, or stroke appeared [19].

### 2.3. Blood Samples

Samples were obtained within the first 12 h after ED arrival. Peripheral venous blood was drawn from patients and kept in dipotassium ethylenediaminetetraacetic acid (EDTA-K2) tubes. Haemolysed blood samples were excluded. Plasma EDTA-K2 was obtained by centrifuging the haemogram tube at 3000 rpm for 10 min; it was subsequently frozen at −80 °C until analysis. To enhance measurement precision, each biomarker was meticulously analysed in duplicate, with all intra-assay coefficients of variation below 10%.

### 2.4. Biomarker Measurement

CRP and PCT were analysed as part of the routine diagnostic work-up. PCT was measured by Test Elecsys BRAHMS PCT in Cobas^®^ 8000 analyser e602 module (Roche Diagnostics GmbH, Sandhofer Strasse, Manheim, Germany); measurement interval: 0.02–100 ng/mL. CRP was measured by Test C-Reactive Protein Gen.3 Cobas^®^ 8000 analyser e701/702 module (Roche Diagnostics GmbH, Sandhofer Strasse, Manheim, Germany); Measurement interval: 0.03–35 mg/dL. Samples above the measurement range were diluted per the manufacturer’s instructions.

In a second analysis, and according to the manufacturer’s directions (TRACE technology [Time-Resolved Amplified Cryptate Emission], MR-proADM and CT-proET-1 were measured with immunofluorescent assays in the BRAHMS KRYPTOR Compact Plus analyser [ThermoFisher Scientific], Waltham, MA, USA). The same reagent kit lots were used on all samples.

For the purpose of this study, biomarkers under examination were dichotomised based on whether they surpassed the third quartile value across the various time points and outcomes considered. Appendix A displays all values for the 75th percentile for each biomarker based on the analysis time.

### 2.5. Statistical Analysis

The statistical analysis was executed using IBM SPSS Statics version 26.0 software, RStudio version 4.2.3 and GraphPad Prism 8.0. Data was summarised as median (1st quartile, 3rd quartile) for continuous variables or count (%) for categorical variables. Statistical significance was considered if *p*-value ˂ 0.05.

The normality of quantitative variables was assessed using the Shapiro-Wilk test. To compare biomarker levels on day 1, day 5, and day 30, either the ANOVA or Kruskal-Wallis test was employed accordingly. Post-hoc comparisons were conducted by adjusting with (Bonferroni/Holm).

For the comparison of baseline characteristics, complications and other qualitative variables, a Chi-squared test with N-1 Campbell correction was performed to correct the low frequency of some categories. Univariate associations are expressed as unadjusted Odds ratios (OR) with 95% confidence intervals (CI).

ROC curves and univariate logistic regression were used to analyse the predictive efficacy of the various biomarkers analysed, alongside the PSI score, across different outcomes and time points. Systematic comparisons were made for both the AUCs (area under the curve) and OR, respectively. In these analyses, biomarkers were dichotomised based on whether their values exceeded the third quartile.

## 3. Results

### 3.1. Patient Characteristics and Clinical Outcomes

In this study, 515 patients were enrolled for biomarker analysis on day 1. At days 5 and 30, 285 and 280 of patients were analysed, respectively (Table 1). Table 1 describes baseline characteristics, comorbidities, and initial severity, as well as analysed outcomes of enrolled patients.

Of all the patients analysed during hospitalisation on day 1 and/or day 5 (n = 529), 63 (11.9%) developed a CVE and 22 (4.2%) died whilst admitted. Of all the patients analysed since hospital discharge until 1-year follow-up (n = 280), 30 (10.7%) developed a CVE and 8 (2.9%) died in the long term.

### 3.2. Endothelial and Inflammatory Markers

Figure 1 depicts biomarker levels in the whole cohort on days 1, 5, and 30. CRP and PCT levels are significantly higher at day 1 (median 17.1 mg/dL and 2.2 ng/mL, respectively) compared to day 5 of hospitalisation (median 7.1 mg/dL and 0.8 ng/mL, respectively) and both of the prior measurements, higher compared to that taken at day 30 (median 0.3 mg/dL and 0.04 ng/mL, respectively). Endothelial injury biomarkers (MR-proADM and CT-proET-1) levels were also significantly more elevated at day 1 (median 1.0 nmol/L and 97.5 pmol/L, respectively) compared to day 5 (median 0.8 nmol/L and 68.6 pmol/L, respectively) and both of the prior measurements, higher compared to the analytical determination at day 30 (median 0.7 nmol/L and 63.3 pmol/L, respectively).

### 3.3. Outcomes

Figure 2 illustrates biomarker levels based on the presence or absence of the considered outcome. On day 1, biomarkers for endothelial damage exhibited significantly higher levels in patients admitted to ICU or those who experienced in-hospital CVE and in-hospital mortality versus those who did not. CRP and PCT only showed significantly higher levels for ICU admission; no significant difference was observed in levels for in-hospital CVE and in-hospital mortality.

On day 5, elevated plasma levels of endothelial biomarkers were observed for in-hospital CVE and in-hospital death compared to patients who had not experienced this event. Concerning inflammatory biomarkers, heightened PCT levels were noted for in-hospital CVE and in-hospital mortality. However, no significant differences were observed in CRP levels for either of the mentioned outcomes.

At day 30, only endothelial biomarkers (MR-proADM and CT-proET-1) exhibited significantly higher levels for CVE at 1-year follow-up; no differences were observed for mortality at 1-year follow-up. In contrast, no significant differences were found in CRP and PCT levels.

Figure 3 and Appendix A outline the OR and 95% CI for biomarker levels at day 1, day 5 and day 30. There was a statistically significant association between MR-proADM and CT-proET-1 and in-hospital CVE, ICU admission and in-hospital mortality on day 1 and day 5. Both biomarkers at day 30 maintained a strong association with CVE at 1-year follow-up (MR-proADM OR, 6.7 [3.0–15.0; *p* < 0.001]; CT-proET-1 OR, 5.0 [2.3–10.9]; *p* < 0.001). PSI-score on day 1 was associated with all outcomes and mainly with in-hospital death (OR, 13.0 [4.6–36.8; *p* < 0.001]). Both classical biomarkers (CRP and PCT) were only associated with ICU admission on day 1. CRP only exhibited association with in-hospital CVE on day 5 (OR, 2.1 [1.2–3.7; *p* = 0.012]); there was no observed association between any of the proposed outcomes on day 30. With respect to PCT, no association was reported for any of the outcomes proposed on day 5 or day 30 of measurement.

To analyse whether the elevation of endothelial damage biomarkers is due to other intercurrent variables, such as some comorbidities, the same analysis was performed and adjusted for diabetes, chronic heart disease, arterial hypertension, and chronic renal failure. Consistent results were obtained with those presented in the total population, showing that the findings are not significantly modified by the presence of these comorbidities in our cohort.

Figure 4 and Appendix A display a comparison of the areas under the ROC curve (AUCs) for each outcome and biomarker. The AUC of endothelial injury biomarkers were superior to that of classic biomarkers for all of the outcomes considered, attaining an AUC similar to the validated PSI score on day 1.

With MR-proADM as the biomarker with the highest AUC and OR and values separated into quartiles (0–25th, 25–50th, 50–75th and >75th percentile), we found a higher percentage of complications (CVE and mortality) as the quartile degree increased (see Figure 5).

## 4. Discussion

The main findings of the study are: (1) High levels of endothelial damage biomarkers (MR-proADM and CT-proET-1) at day 1 and day 5 are associated with ICU admission, in-hospital CVE and in-hospital mortality; (2) At day 30, MR-proADM and CT-proET-1 maintains an association with the occurrence of CVE until 1-year follow-up; (3) Endothelial injury biomarkers outperform classic inflammatory biomarkers (CRP and PCT) exhibiting superior value of OR and AUC for all of the outcomes considered; (4) MR-proADM presents as the most optimal biomarker in terms of being associated with both short and long term complications in the analysed outcomes.

Examining traditional biomarkers in CAP and COVID-19 has been widely performed. Nevertheless, emerging biomarkers, such as endothelial markers, are starting to become more relevant in routine clinical practice. In CAP, the main potential use of endothelial damage biomarkers (MR-proADM and CT-proET-1) is for short and long-term survival [14,18,20]. However, there is scarce literature on biomarker measurements for endothelial damage during follow-up after hospital discharge [19].

Our study confirms initially elevated levels of the biomarkers. These, thereafter, exhibit a progressively declining kinetics pattern by day 5 and biomarker levels reach the normal range by day 30, except for a small percentage of patients who continue to display slightly elevated levels.

Initial, higher levels of MR-proADM and CT-proET-1 (day 1) were found in patients developing CVE during hospitalisation, who later required ICU admission or experienced death. However, CRP and PCT presented only significantly higher levels in those requiring ICU admission. In fact, OR for CVE was reported as better for MR-proADM and CT-proET-1. They were only surpassed by the PSI score for in-hospital mortality; this was not the case, though, for all the other outcomes analysed. Concerning in-hospital CVE or in-hospital mortality outcomes, our study shows that plasma levels of MR-proADM equal to or greater than 1.543 nmol/L measured at day 1 (equivalent to ≥75th percentile) exhibit a higher percentage of events compared to those patients with lower plasma levels. These data are consistent with other publications, which propose a plasma MR-proADM value between 1.3 and 1.6 nmol/L as the optimal cut-off point for predicting death [21,22]. For other in-hospital complications, Lacoma et al. propose a cut-off between 0.95 and 1.5 nmol/L for the onset of complications related to CAP; they do not define a cut-off point for CVE [23].

Although a reduction in biomarker plasma levels was observed on day 5, results for the association between biomarker measurement and outcomes were similar to those depicted on day 1. Therefore, in the absence of a determination at admission, it could be requested during hospitalisation.

Our findings reveal that at day 30, which aligns with the routine follow-up medical visit, endothelial biomarkers maintain a significant association with the onset of CVE at 1-year follow-up and show no relationship with mortality at 1-year follow-up. In contrast, this association was not observed for inflammatory biomarkers. Pneumonia is well known to increase cardiovascular risk in the short and long term, lasting for at least 10 years [3]. Nevertheless, there is scarce literature on biomarker measurements during extended follow-up periods after hospital discharge and their association with complication onset. To our knowledge, there is only one study available that measures endothelial damage biomarkers during follow-up and aims to explore the potential association with the onset of long-term CVE [19]. Menéndez et al. showed that measurements of both interleukin-6 (IL-6) and endothelial damage biomarkers (pro-ET-1 or proADM) are associated with a high risk of CVE in the long term. Our results confirm the usefulness of endothelial damage biomarkers at day 30 in predicting CVE in the long term as well. Furthermore, our study shows that the employment of MR-proADM alone is sufficient and useful.

Classic biomarkers (CRP and PCT) used in current algorithms for CAP have been valuable and relevant [24]. Our study confirms that they are, indeed, helpful in the acute phase; however, including the use of endothelial biomarkers in conjunction with CRP and PCT represents an added value in clinical practice, giving a statistically stronger association for both the acute and chronic phases, and highlight potential cardiovascular complications.

Additionally, our research demonstrates that MR-proADM and CT-proET-1 are both excellent markers of worse prognosis during hospitalisation for CAP, highlighting their utility in predicting CVE and at day 30 for CVE. Nevertheless, the integration of MR-proADM in clinical practice may be more feasible than that of CT-proET-1. Reasons include the availability of a commercial MR-proADM kit approved for healthcare use, as well as experience in other pathologies such as sepsis [15]. CT-proET-1 is currently more limited to the area of research. The availability of devices in which these automated kits can be implanted makes it possible to incorporate them into any healthcare laboratory. Furthermore, measuring MR-proADM is a straightforward process: it can be rapidly analysed with self-analysers that boast a high performance of samples (according to technical specifications: 60 samples/1 h).

In this context, MR-proADM could be envisioned as a single tool to stratify patients that may present CVE or a worsening during the acute phase or even up to one year after the event, regardless of CAP aetiology. This makes MR-proADM a solid biomarker to assess both acute damage and follow-up after discharge, and it creates the possibility of establishing closer follow-up protocols in patients with a higher risk of presenting cardiovascular complications.

Some limitations of our study must be acknowledged: (1) There is no cohort of healthy controls for comparison in our study. Despite this, the median value and interquartile range for both endothelial damage biomarkers have been established in previous studies, showing significantly lower levels than those found in our patients (MR-proADM: 0.4 [0.4–0.5] nmol/L in 1298 healthy individuals; CT-proET-1: 44.3 [10.5–77.4] pmol/L in 326 healthy individuals) [25,26]; (2) Due to the absence of biomarker data prior to the current pneumonia episode, it is difficult to clarify which degree is due to the acute process and which one is due to the pre-existing endothelial injury secondary to ageing and chronic diseases such as diabetes or arterial hypertension among others [27]. The possible presence of previous endothelial damage, which is difficult to quantify, increases the vulnerability of patients. This makes them more susceptible to suffering further endothelial injury from pneumonia and, consequently, systemic involvement. In any case, the existence of either possibly previous endothelial damage or endothelial damage secondary to CAP predisposes patients to a worse outcome. (3) There is a decrease in sample size at follow-up (day 1, n = 515 vs. day 30, n = 280). Although sufficient statistical power was obtained for long-term CVE, this was not the case for less common events such as 1-year mortality. Further research is needed to clarify this point.

## 5. Conclusions

Our findings provide relevant information on endothelial damage biomarkers and prognosis in CAP. When compared to classic inflammatory biomarkers, endothelial damage biomarkers show a stronger association with a worse prognosis. MR-proADM is the most optimal biomarker to predict short- and long-term complications in CAP. The study reveals results that are highly applicable to clinical practice. Incorporating measurements of MR-proADM into clinical laboratories may provide extremely valuable information when it comes to managing patients with CAP. Due to the high association between elevated MR-proADM and in-hospital mortality, CVE and ICU admission in the short term, and CVE in the long term, routine implementation of this biomarker in the follow-up of patients with a previous episode of CAP may allow for a closer, stronger follow-up protocol.

## Figures and Tables

**Figure 1 biomedicines-12-02413-f001:**
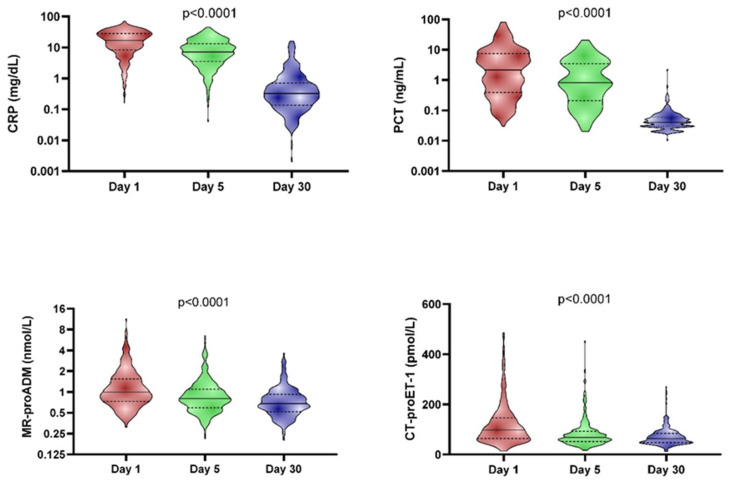
Differences between biomarker levels at day 1, day 5 and day 30. The differences among all groups were statistically significant (*p* < 0.0001). The solid line represents the median, and the dashed lines represent the interquartile range.

**Figure 2 biomedicines-12-02413-f002:**
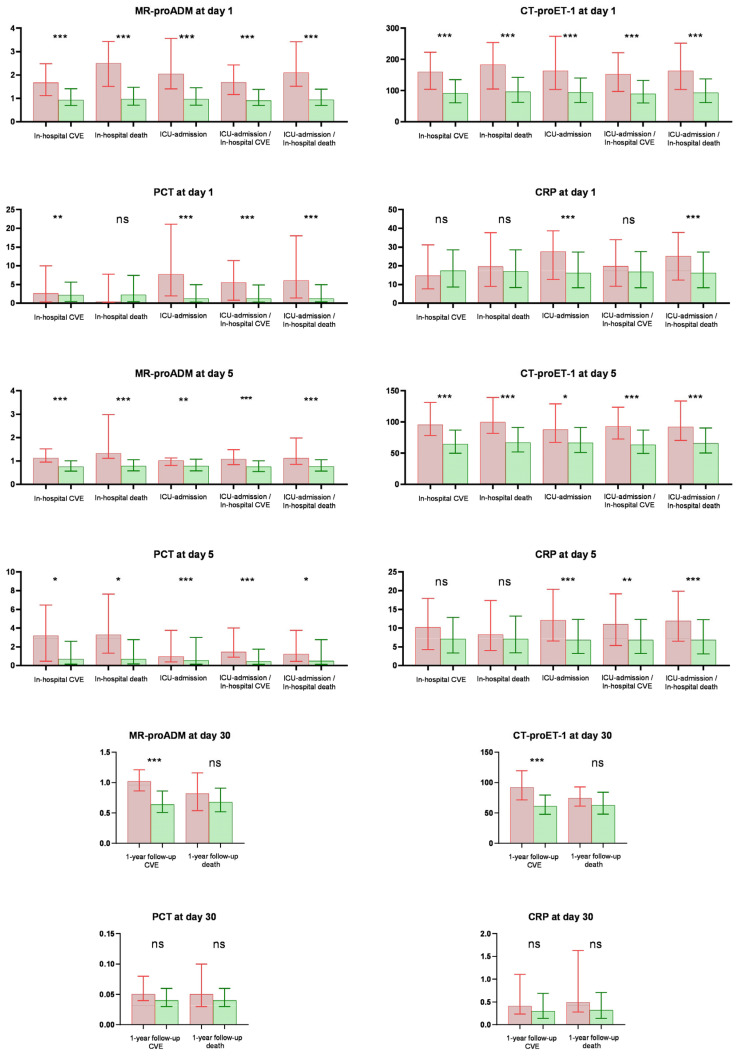
Differences in biomarker levels were measured across various days based on the analysed outcome. ***: <0.001; **: <0.01; *: <0.05; ns: not significant. Box represent the median value, and whiskers represent the interquartile range. Red colour: the presence of outcome. Green colour: absence of outcome.

**Figure 3 biomedicines-12-02413-f003:**
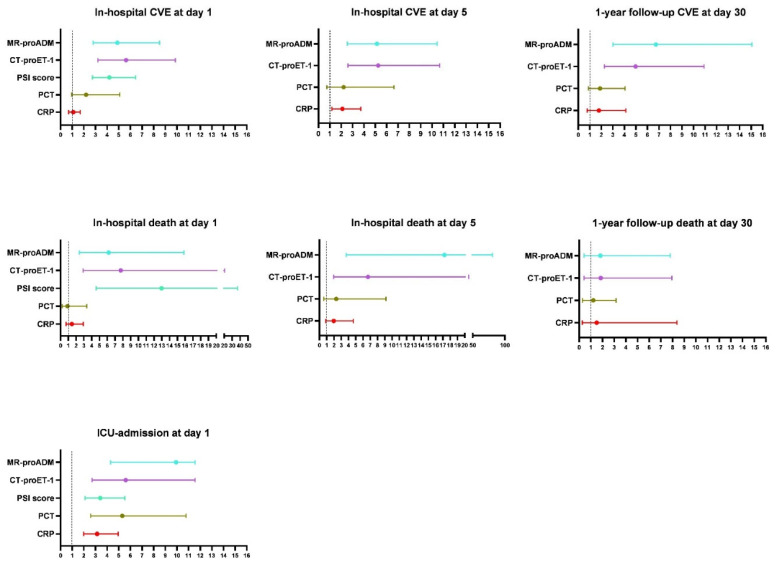
The relationship between biomarkers/PSI score and outcomes across different day measurements is expressed in OR. Circles represent Odds Ratio (OR) and whiskers, 95% confidence intervals. Blue represents MR-proADM, purple represents CT-proET-1, light green represents PSI score, dark green colour represents PCT, and red colour represents CRP.

**Figure 4 biomedicines-12-02413-f004:**
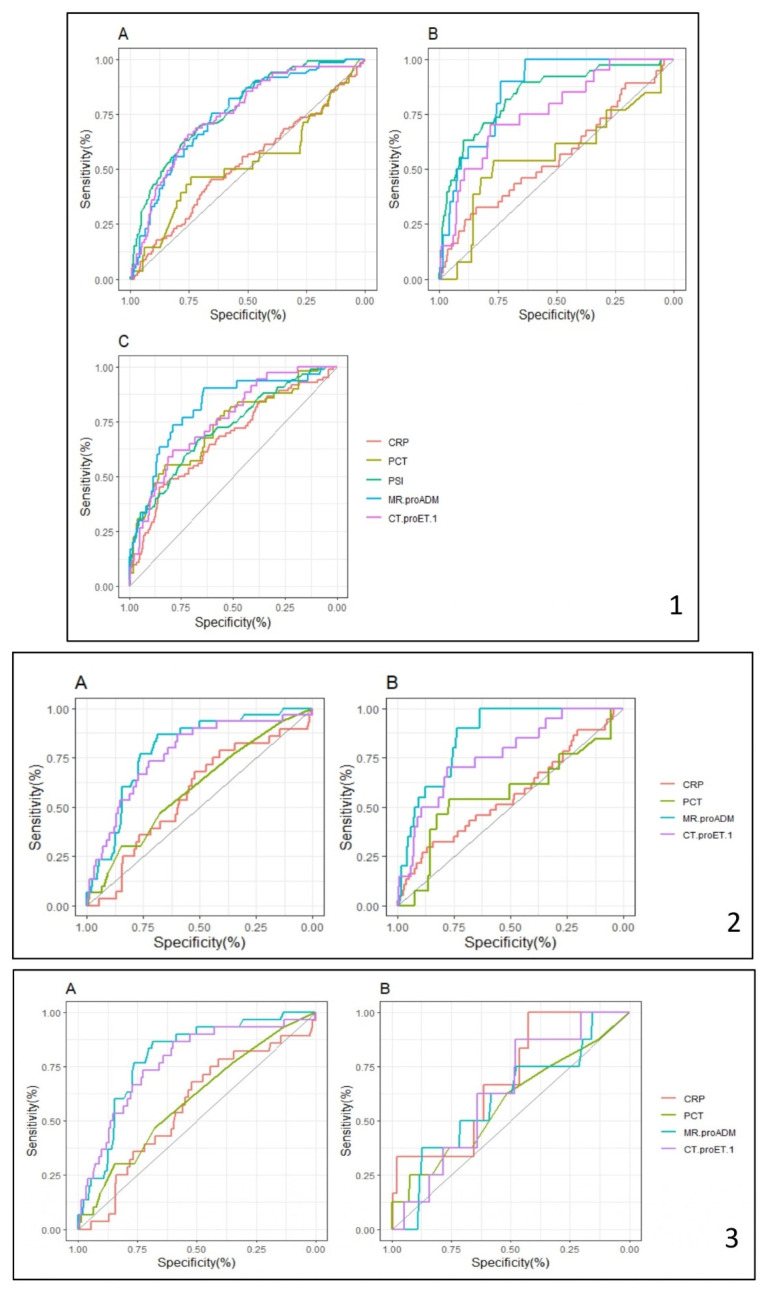
ROC curves for diagnostic accuracy of biomarkers/PSI score in the different outcomes evaluated. (**1**) = Day 1; (**2**) = Day 5; (**3**) = Day 30; (**1A**,**2A**) = In-hospital CVE; (**1B**,**2B**) = In-hospital mortality; (**1C**) = ICU admission; (**3A**) = 1-year follow-up CVE; (**3B**) = Mortality at 1-year follow-up.

**Figure 5 biomedicines-12-02413-f005:**
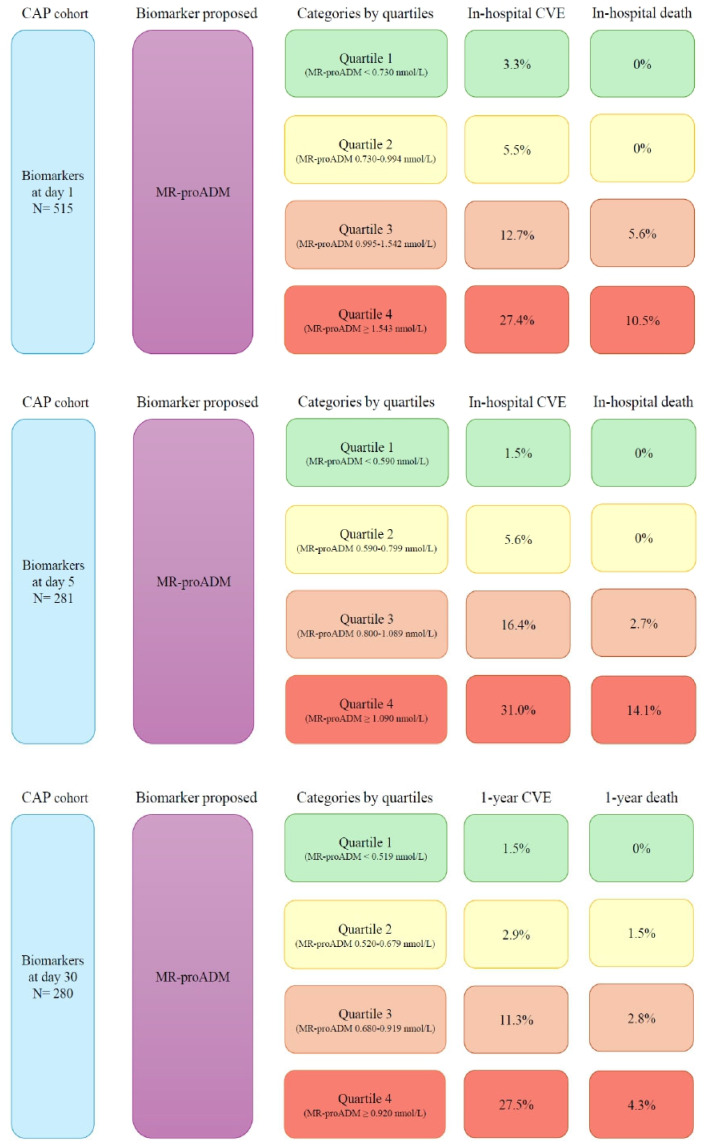
Higher percentage of complications as the quartile degree of MR-proADM increases.

**Table 1 biomedicines-12-02413-t001:** Baseline characteristics, severity, and respiratory support.

	Patients with Biomarkers at Day 1 (*n* = 515)	Patients with Biomarkers at Day 5 (*n* = 285)	Patients with Biomarkers at Day 30 (*n* = 280)
**Age, years, median (IQR)**	71 (59, 80)	72 (60, 81)	71 (58, 79)
**Male sex, no. (%)**	326 (63.3)	189 (66.3)	175 (62.5)
**Current or former smokers, no. (%)**	333 (64.7)	170 (59.6)	158 (56.4)
**Co-existing conditions, no. (%)**			
**HBP**	274 (53.2)	166 (58.2)	150 (53.4)
**Diabetes**	132 (25.6)	78 (27.4)	68 (24.3)
**Overweight ***	117 (22.7)	53 (18.6)	45 (16.1)
**COPD**	132 (25.6)	85 (29.8)	69 (24.6)
**Asthma**	43 (8.3)	22 (7.7)	25 (8.9)
**Chronic heart disease**	185 (35.9)	109 (38.2)	95 (33.9)
**Chronic renal disease**	69 (13.4)	38 (13.3)	28 (10.0)
**Neurologic disease**	81 (15.7)	48 (16.8)	38 (13.6)
**Severity**			
**SpO2/FiO2 at admission, median (IQR)**	443.8 (419.1, 457.1)	443.3 (416.0, 457.1)	447.6 (430.6, 457.1)
**PSI score, no (%)**			
**PSI 1–3**	295 (57.3)	155 (54.4)	175 (62.5)
**PSI 4–5**	220 (42.7)	130 (45.6)	105 (37.5)
**In- hospital clinical outcomes, no (%)**			
**In-hospital CVE**	62 (12.0)	39 (13.7)	NA
**In-hospital death**	20 (3.9)	12 (4.2)	NA
**ICU admission**	34 (6.6)	NA	NA
**Follow-up clinical outcomes, no (%)**			
**1-year follow-up CVE**	75 (14.6)	43 (15.1)	30 (10.7)
**1-year follow-up death**	32 (6.2)	14 (4.9)	8 (2.9)

IQR: interquartile range; HBP: high blood pressure; COPD: chronic obstructive pulmonary disease; CVE: cardiovascular events; SpO2/FiO2: peripheral blood oxygen saturation/fraction of inspired oxygen; PSI: pneumonia severity index; ICU: intensive care unit; NA: not applicable. * Body mass index ≥25.

## Data Availability

The datasets used and/or analysed during the current study are available from the corresponding author upon reasonable request.

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
