# Peer review of "Endothelial Biomarkers Are Superior to Classic Inflammatory Biomarkers in Community-Acquired Pneumonia"

_biomedicines, 2024, doi:10.3390/biomedicines12102413_

Round 1

Reviewer 1 Report

Comments and Suggestions for Authors

The paper deals with the possible application of endothelial damage biomarkers in community-acquired pneumonia. The results are of interest mainly for pro-ADM usefulness in the clinical practice.

I have only minor points to be addressed

Figure 1: which parameters are reported in the plot diagrams (median, CI, ....)?

Were there patients evolving in sepsis/septic shock or were they excluded from the prospective analysis?

Why Kaplan-Meier curves have not be used in the assessment of biomarkers contribution to different outcomes?

Author Response

The paper deals with the possible application of endothelial damage biomarkers in community-acquired pneumonia. The results are of interest mainly for pro-ADM usefulness in the clinical practice.
I have only minor points to be addressed
1. Figure 1: which parameters are reported in the plot diagrams (median, CI, ....)?
- We have added in the manuscript that solid line represents the median and dashed lines represent the interquartile range.
2. Were there patients evolving in sepsis/septic shock or were they excluded from the
prospective analysis?
- Patients with sepsis/septic shock were not excluded. Patients with CAP of any severity were included, including septic shock, sepsis, or need for ICU admission due to organ failure.
3. Why Kaplan-Meier curves have not be used in the assessment of biomarkers contribution
to different outcomes?
- Unfortunately, the exact date of the events was not recorded. We have data on whether events occurred in-hospital or out-of-hospital until the follow-up year.

Reviewer 2 Report

Comments and Suggestions for Authors

The manuscript by Paula González-Jiménez and co-workers describes a comprehensive analysis of endothelial damage biomarkers compared to traditional inflammatory biomarkers in predicting outcomes for patients with community-acquired pneumonia. The authors have performed a thorough investigation of both short-term and long-term outcomes, including ICU admission, cardiovascular events, and mortality. The results are important and deserve publication after some improvements in the manuscript.

In Table 1, the authors presented several subgroups of the enrolled patients (men/women, smokers/non-smokers, etc.). Were there statistically significant differences between these subgroups regarding the biomarkers investigated? These issues should be addressed in the manuscript.

Figure 2:  Some symbols in the pictures are too small and should be enlarged.

The captions of the supplementary tables are given under the corresponding tables. Please place them before the tables.

I recommend major revision of the manuscript before acceptance.

Reviewer 3 Report

Comments and Suggestions for Authors

This manuscript evaluates the utility of endothelial damage biomarkers (CT-proET-1 and MR-proADM) compared to classical inflammatory markers (CRP and PCT) in predicting short- and long-term outcomes in patients with community-acquired pneumonia (CAP). The study presents novel insights into the role of endothelial biomarkers in predicting cardiovascular events (CVE), ICU admissions, and mortality in both acute and long-term settings. The findings suggest that MR-proADM, in particular, is a superior predictor for complications related to CAP.

The study addresses an important clinical challenge, which is the management of patients with CAP, particularly in predicting short- and long-term complications like cardiovascular events (CVE).
The study employs a well-defined prospective design, with clear inclusion/exclusion criteria, detailed statistical analysis, and appropriate follow-up to assess both short- and long-term outcomes. The use of ROC curve analysis and univariate logistic regression strengthens the validity of the findings.

The study compares classical biomarkers (CRP and PCT) with endothelial markers (MR-proADM and CT-proET-1), offering a comprehensive view of their respective predictive capabilities. This allows for a meaningful comparison and adds depth to the analysis.

Major/minor comments

The manuscript acknowledges that elevated endothelial biomarkers could be partially due to pre-existing endothelial damage related to chronic conditions such as hypertension, diabetes, or aging. However, there is insufficient discussion on how these confounding factors were controlled in the analysis or how they could have impacted the results.

While the sample size at day 1 is robust (n=515), it decreases at follow-up (n=280 at day 30). This drop in sample size, while understandable, may impact the statistical power of long-term outcome analyses, especially for rare events such as mortality at 1-year follow-up.

The title should be revised. The phrasing "Is MR-proADM The New CRP?" may be overly suggestive, as CRP and MR-proADM serve different purposes. Consider revising to a more neutral title.

The abstract could benefit from specifying the statistical significance of the findings (e.g., provide p-values).
The methods section is detailed and transparent, providing sufficient information for replication. The use of standard assays for biomarker measurements is commendable, though additional information on how potential confounders (e.g., pre-existing conditions) were controlled would strengthen this section.

The study provides valuable insights into the role of endothelial biomarkers in CAP, particularly MR-proADM, but there are a few areas that need further clarification or expansion:

  • Address the potential impact of pre-existing conditions on biomarker levels.
  • Consider adding a brief discussion on the implications of the reduced sample size at follow-up.

The conclusion is appropriate, emphasizing the potential clinical utility of MR-proADM as a predictor for CAP complications. The authors make a compelling case for incorporating MR-proADM into routine clinical practice for managing CAP patients.

Note:

 ^ Senior authors? You mean equal contributions?

Author Response

This manuscript evaluates the utility of endothelial damage biomarkers (CT-proET-1 and MR-proADM) compared to classical inflammatory markers (CRP and PCT) in predicting short- and long-term outcomes in patients with community-acquired pneumonia (CAP). The study presents novel insights into the role of endothelial biomarkers in predicting cardiovascular events (CVE), ICU admissions, and mortality in both acute and long-term settings. The findings suggest that MR-proADM, in particular, is a superior predictor for complications related to CAP.
The study addresses an important clinical challenge, which is the management of patients with CAP, particularly in predicting short- and long-term complications like cardiovascular events (CVE).

The study employs a well-defined prospective design, with clear inclusion/exclusion criteria, detailed statistical analysis, and appropriate follow-up to assess both short- and long-term outcomes. The use of ROC curve analysis and univariate logistic regression strengthens the validity of the findings.
The study compares classical biomarkers (CRP and PCT) with endothelial markers (MR-proADM and CT-proET-1), offering a comprehensive view of their respective predictive capabilities. This allows for a meaningful comparison and adds depth to the analysis.
Major/minor comments
1. The manuscript acknowledges that elevated endothelial biomarkers could be partially due to pre-existing endothelial damage related to chronic conditions such as hypertension, diabetes, or aging. However, there is insufficient discussion on how these confounding factors were controlled in the analysis or how they could have impacted the results.

- To analyze whether the elevation of endothelial damage biomarkers is due to other intercurrent variables such as some comorbidities, the same analysis was performed adjusted for diabetes, chronic heart disease, arterial hypertension and chronic renal failure. Consistent results were obtained with those presented of total population, showing that the findings are not significantly modified by the presence of these comorbidities. These results are not included in the manuscript because of similar data, but we have added an explanation in the Results section (3.3. Outcomes).
- As suggested, in the new version, we have expanded the discussion in the limitations section.

2. While the sample size at day 1 is robust (n=515), it decreases at follow-up (n=280 at day 30). This drop in sample size, while understandable, may impact the statistical power of long-term outcome analyses, especially for rare events such as mortality at 1-year follow-up.
- As the reviewer rightly points out, this may impact the statistical power. However, they have been losses due to the follow-up (e.g., discharge before 5 days, no 30 days follow- up visits).
- We have added this observation in potential limitations: “There is a decrease in sample size at follow-up (day 1, n=515 vs. day 30, n=280). Although sufficient statistical power was obtained for long-term CVE, this is uncertain for less common events such as 1-year
mortality. Further research is needed to clarify this point.”

3. The title should be revised. The phrasing "Is MR-proADM The New CRP?" may be overly suggestive, as CRP and MR-proADM serve different purposes. Consider revising to a more neutral title.
- In the new version, as suggested, we have removed "Is MR-proADM the new CRP?" from the title to make it more neutral.

4. The abstract could benefit from specifying the statistical significance of the findings (e.g., provide p-values).
- We agree with your assessment, however, due to the limitation of 250 words in the abstract, we have not been able to include specific p-value data.

5. The methods section is detailed and transparent, providing sufficient information for replication.
- Thank you very much for your kind appreciation.

6. The use of standard assays for biomarker measurements is commendable, though additional information on how potential confounders (e.g., pre-existing conditions) were controlled would strengthen this section.

- Biomarkers were analyzed in subgroups (e.g. diabetes, chronic heart disease, arterial hypertension and chronic renal failure), but no statistically significant differences were found, so we decided not to include them in the manuscript. For this reason, it was
included only biomarkers analysis with the total population. We have explained this point in the new version.

7. The study provides valuable insights into the role of endothelial biomarkers in CAP, particularly MR-proADM, but there are a few areas that need further clarification or expansion:
• Address the potential impact of pre-existing conditions on biomarker levels.
• Consider adding a brief discussion on the implications of the reduced sample size at follow-up.
We have added these assessments in potencial limitations (discussion section).

The conclusion is appropriate, emphasizing the potential clinical utility of MR-proADM as a predictor for CAP complications. The authors make a compelling case for incorporating MR- proADM into routine clinical practice for managing CAP patients.

8. Note: ^ Senior authors? You mean equal contributions?
- Rosario Menéndez and Raúl Méndez have been identified as senior authors as they had equal contribution in the revision and supervision of the manuscript. In the new version, we have changed ‘Senior authors’ for ‘Equal contribution

Reviewer 4 Report

Comments and Suggestions for Authors

Based on clinical studies, the authors propose to use endothelial markers CT-proET-1 and MR-proADM as indicators of worse outcomes in patients with community-acquired pneumonia in both the short and long term, especially in case of cardiovascular complications.

Please discuss the diseases and conditions of patients associated with elevated endothelial markers more broadly in the Discussion section. 

Please discuss how likely it is in a group of relatively healthy people without CAP that markers could be elevated for other reasons?

Please decipher the abbreviation ICU.

Author Response

Based on clinical studies, the authors propose to use endothelial markers CT-proET-1 and MR-proADM as indicators of worse outcomes in patients with community-acquired pneumonia in both the short and long term, especially in case of cardiovascular complications.

1. Please discuss the diseases and conditions of patients associated with elevated endothelial markers more broadly in the Discussion section.
- To analyze whether the elevation of endothelial damage biomarkers is due to other intercurrent variables such as some comorbidities, the same analysis was performed adjusted for diabetes, chronic heart disease, arterial hypertension and chronic renal failure. Consistent results were obtained with those presented of total population, showing that the findings are not significantly modified by the presence of these comorbidities. These results are not included in the manuscript because of similar data, but we have added more information in the Results section.

- As suggested, in the new version, we have expanded the discussion in the limitations
section.

2. Please discuss how likely it is in a group of relatively healthy people without CAP that markers could be elevated for other reasons?
- Unfortunately, we do not have data analyzed in healthy controls, which we have reflected in the limitations section (discussion). After a review of the literature, the values of biomarkers of endothelial damage (MR-proADM and CT-proET-1) are lower than those obtained in our patients with CAP (line 311-316).

3. Please decipher the abbreviation ICU.
- We have properly deciphered the abbreviation ICU in the manuscript

Round 2

Reviewer 2 Report

Comments and Suggestions for Authors

The revised version of the manuscript was improved significantly by the authors. All the comments have been addressed. I recommend acceptance of the manuscript for publication in the revised form.